# Effect of Linseeds and Hemp Seeds on Milk Production, Energy and Nitrogen Balance, and Methane Emissions in the Dairy Goat

**DOI:** 10.3390/ani11092717

**Published:** 2021-09-17

**Authors:** Luca Rapetti, Stefania Colombini, Giovanna Battelli, Bianca Castiglioni, Federica Turri, Gianluca Galassi, Marco Battelli, Gianni Matteo Crovetto

**Affiliations:** 1Dipartimento di Scienze Agrarie e Ambientali—Produzione, Territorio, Agroenergia, Università degli Studi di Milano, via Celoria 2, 20133 Milano, Italy; luca.rapetti@unimi.it (L.R.); stefania.colombini@unimi.it (S.C.); gianluca.galassi@unimi.it (G.G.); marco.battelli@unimi.it (M.B.); 2Istituto di Scienze delle Produzioni Alimentari, Consiglio Nazionale delle Ricerche, via Celoria 2, 20133 Milano, Italy; giovanna.battelli@ispa.cnr.it; 3Istituto di Biologia e Biotecnologia Agraria, Consiglio Nazionale delle Ricerche, via Einstein, 26900 Lodi, Italy; castiglioni@ibba.cnr.it (B.C.); federica.turri@ibba.cnr.it (F.T.)

**Keywords:** goat, milk fatty acids, linseed, hemp seed, methane

## Abstract

**Simple Summary:**

The inclusion of whole oilseeds in the diets of ruminants can be a useful strategy for reducing methane emissions and improving milk quality. This study evaluated the effects of the inclusion of whole hemp seeds or linseeds in the diet of dairy goats. The results showed that neither seed caused a reduction in methane emission or an increase in milk yield, but both seeds improved the milk quality in terms of fatty acid composition.

**Abstract:**

The effect of whole linseeds or hemp seeds on milk production, energy and nitrogen balance, and methane emission was studied in 12 Alpine goats using respiration chambers. Diets tested were a control diet (C) and two diets supplemented with whole linseeds (L) or hemp seeds (H) at 9.3% on a dry matter (DM) basis. DM intake was similar among treatments, whereas DM and organic matter digestibility were lower for L compared to C. Milk yield (2.30 kg/d on average) and rumen fermentation profile were not affected by treatments. Treatment also did not affect the milk composition, with the exception of fat, which was higher in H and L compared to C (4.21, 3.94, and 3.20%, respectively). Oilseed supplementation caused a reduction in the concentration of de novo fatty acids (FA) (41.1, 48.8, and 64.1% of FA, for L, H, and C, respectively). Moreover, L and H diets reduced the sum of saturated FA, and increased monounsaturated FA, whereas only the L diet increased the concentration of polyunsaturated FA. Regarding methane production, and nitrogen and energy balances, no differences were registered among the diets. Our research indicates that including whole linseeds and hemp seeds in the dairy goat diet is an effective strategy for increasing milk fat content and positively modifying the milk FA composition, without a change in nitrogen and energy balances, but also without a reduction in enteric methane emission.

## 1. Introduction

Feeding animals with sources of polyunsaturated fatty acids (PUFA) has been of interest to enhance the concentration of beneficial fatty acids (FA) in animal products, specifically n-3 PUFA, associated with positive effects on human health [1]. The physical form of the lipids (i.e., seeds vs. oil), the level of inclusion in the diet, and the interaction with other dietary ingredients are among the main factors affecting milk fatty acid composition and concentration [2]. For example, Luna et al. [3] showed that feeding both whole linseed and sunflower oil to lactating goats changed the fatty acid composition of milk with a noticeable increase in the secretion of α-linolenic, rumenic, and vaccenic acids. Moreover, the expected increase in milk fat content due to lipid supplementation could help solve the technological problems of the goat cheese industry related to low milk fat content, especially when the fat content falls below the protein content [4].

Supplementation of diets with lipids, such as linseeds, is a dietary strategy recognized to lower enteric CH_4_ emissions [5,6,7]. Added fats decrease CH_4_ emissions by lowering the quantity of organic matter fermented in the rumen, the activity of ruminal methanogens, and protozoal numbers. Moreover, for lipids rich in unsaturated fatty acids, rumen CH_4_ production is also reduced for the biohydrogenation of FA [8]. In this regard, including lipids in the diet as oilseeds rather than oil limits the extent of ruminal biohydrogenation of PUFA because seed hulls restrict the access of bacterial lipases to storage triacylglycerol [9].

Oilseeds commonly used in ruminant diets include sunflower seed, linseed, rapeseed, and soybean seed [2]. However, as underlined by Wang et al. [10], the list of possible alternatives is much longer. Examples of unconventional oilseeds rich in PUFA are safflower, poppy, hemp, and camelina. However, few in vivo studies have described the effects of these alternative seeds fed to ruminants [10].

Recently there has been a renewed interest in a nondrug type of *Cannabis sativa* L. supported by the dissemination of studies and data [11,12] that highlight the beneficial properties of their fruits (hemp seeds). Since 1990, dozens of countries have authorized the licensed growth and processing of hemp cultivars with relatively low levels of tetrahydrocannabinol [13]. In Europe and North America, hemp seeds for food were rediscovered in the mid-1990s with the reintroduction of hemp as a source of technical fiber [14]. The hemp varieties allowed for cultivation in Europe must not exceed 0.2% tetrahydrocannabinol (THC) [15] and hemp seeds are practically free of THC (maximum 12 mg THC/kg) [15]. Hemp seeds typically contain over 300 g of oil/kg, about 250 g of protein/kg, and considerable amounts of dietary fiber, vitamins, and minerals [16]. In addition, hemp seed protein has a favorable amino acid profile and resists degradation in the rumen, yet is highly digestible in the total gastrointestinal tract [17]. Hemp seed oil is high in linoleic and linolenic acids: 60% and 19.3% of total FA, respectively [18].

To the best of our knowledge, no studies have tested whole hemp seeds on lactating goats, and few studies have been conducted on dairy cattle and sheep fed diets supplemented with hemp cakes [17,19,20].

The aim of this study was to evaluate the effects of diets enriched with polyunsaturated fatty acids derived from hemp seeds or linseeds, on milk production, in vivo digestibility, N and energy balance, and CH_4_ emissions in lactating dairy goats.

## 2. Materials and Methods

The study was conducted at the Research Center “Cascina Baciocca” at Cornaredo (Milan) of the Università degli Studi di Milano (Italy). Animal procedures were conducted under the approval of the University of Milan Ethics Committee for animal use and care (authorization 88/14, 29 January 2014), in accordance with the guidelines of the Italian law on animal welfare for experimental animals (Legislative Decree 116/92), in force at the time of the trial.

### 2.1. Animals, Experimental Design and Diets

Twelve multiparous lactating Alpine goats were divided into two groups of six animals, each with animals of similar body weight (BW), milk yield (MY), and days in milk (DIM). Before the beginning of the experiment, when the goats were receiving the same transition diet, BW, MY, and DIM were, on average, 48.5 kg, 2.65 kg/d, and 49 days for the 12 goats. Within each group, the six animals were then randomly allocated in pairs to one of the three following dietary treatments: a control (C) diet, a diet supplemented with linseeds (L), and a diet supplemented with hemp (*Cannabis sativa* L. var. *Futura 75*) seeds (H). The goats were confined in three free stalls on wheat straw bedding according to the treatments.

The three diets, formulated to meet the protein and energy requirements of lactating goats producing 2.8 kg of milk/day, according to the French INRA system [21], were constituted of the same basal ration that accounted for 85.6% of total diet, on a dry matter (DM) basis, and three different concentrate supplements (14.4% on total dietary DM). The basal ration, containing the forages and some concentrate raw materials, was prepared with a mixer wagon allowing a reduction in the forage particle size. The concentrate supplements, prepared by weighing and mixing each ingredient in a small mixer with a capacity of 100 L, were formulated to have a similar nitrogen and net energy content. For this purpose, the amounts of the two types of seeds used in diet L and H were substituted in diet C with the same amount of corn flakes and soybean meal in a ratio of 60:40. The composition of the three different diets, distinguishing for basal ration and concentrate supplements, is reported in Table 1. The control diet was characterized by a higher inclusion of corn flakes (18.6 vs. 13.1% on DM, respectively, for C vs. L and H diets) and soybean meal (6.61 vs. 2.93% on DM, respectively, for C vs. L and H), which were partially substituted by linseed and hemp seeds in L and H diets. Linseed and hemp seeds were included in concentrate supplements at a concentration of about 65%, and the concentrated supplements represented 14.4% of total diet DM. Hence, the level of inclusion of linseeds and hemp seeds was, on average, 9.3% on total DM.

The three concentrate supplements were administered individually twice daily (8.30 a.m. and 5.30 p.m.) about 30 min before the basal ration. The amount offered in the morning and in the afternoon represented 45 and 55% of the total diet, respectively. The orts were recorded once daily, and the feeding rate was adjusted to yield orts on the basis of about 20% of the amount supplied (on an as-fed basis).

The basal ration was administered in the collective manger of each box. During the entire experiment, the goats had free access to water. The animals were adapted to the experimental diets for 30 days before the beginning of the experimental period during which data was recorded. The last three days of feed adaptation, the goats were allocated to individual metabolic cages.

Goats were mechanically milked once daily at 8.00 a.m. from the beginning of lactation.

### 2.2. Digestibility and Respiratory Exchanges Determination

Dietary treatments were simultaneously tested by allocating the goats to individual metabolic cages. Individual data collection for administered diet, orts, feces, urine, and milk lasted six days. During the data collection period, three consecutive days were spent in individual open-circuit respiration chambers to measure three 24 h cycles of respiratory exchanges and methane emission.

The chambers measured 3.6 (length) × 2.4 (width) × 2.3 m (height), and each contained a small pre-chamber for the personnel entrance, and wide glass walls to allow the goats to see each other and outside. In the chambers, the air temperature was maintained at 20 ± 1 °C and a low negative pressure was fixed inside the chambers to prevent loss of air. The air flow through the chambers was measured using a diaphragm flowmeter (PH 20/335 G 25, 40 m^3^/h, Sacofgas, Città di Castello, Perugia, Italy). The air flux was on average maintained at 20 ± 1 m^3^/h. The daily O_2_ consumption and the CO_2_ and CH_4_ production were determined by measuring the volume of air circulating in the system in 24 h (and by referring to the standard temperature and pressure conditions), and multiplying this volume by the difference between the relative concentrations of the gases measured continuously in the ingoing and the outgoing air. The CH_4_ and CO_2_ concentrations were measured using an URAS 4 analyzer (Hartmann and Braun AG, Frankfurt am Main, Germany). The oxygen concentration was measured using a Magnos 6G analyzer (Hartmann and Braun AG, Frankfurt am Main, Germany). The gas concentrations were measured every 450 s, considering 102.5 s of air change and 10 s of O_2_, CO_2_, and CH_4_ determination for each chamber and the external air, for a total of 192 observations/d for each gas and each goat. Corrections were applied to account for the entrance of personnel. Particularly, regarding the calculation of the amount of CH_4_ production, expressed in g per day, it was calculated considering 1 L of CH_4_ is equivalent to 0.71682 g. Moreover, the CH_4_ energy output (kJ/d), was calculated multiplying the determined volume (L/d) of CH_4_ per 39.5388. The total heat production was determined using the Brouwer equation [22]: heat production (kcal/d) = 3.866 O_2_ + 1.200 CO_2_ − 1.431 N − 0.518 CH_4_, where gas volumes (L/d) are expressed at standard conditions and N (g/d) is the urinary N.

Urine and feces were separately collected daily as follows: goats were fitted with Foley urinary catheters (model 1851H18, C. R. Bard Inc., Covington, GA, USA) and urine was collected in plastic bottles containing sulfuric acid (20% vol/vol) to acidify the urine of each goat until pH 2 in order to prevent ammonia losses. Due to the structure of the metabolic cages, the feces were conveyed by gravity into a container. The feces and urine were weighed daily, sampled (20% of the total weight), and pooled per goat during each collection period. Feces, diets, and ort samples were dried in a ventilation oven at 55 °C until constant weight was achieved. After drying, the samples were ground to 1 mm using a Fritsch mill (Pulverisette 19, Fritsch GmbH, Idar-Oberstein, Idar-Oberstein, Germany). A fresh feces subsample was used for N analysis. The N balance was determined by considering the N volatilized in the chamber, which was measured from the N concentration of the water condensed by the air conditioning system. This process involved collecting the total volume of condensed water in plastic canisters containing 20% sulfuric acid (*v*/*v*), which were placed inside of the chambers. The water volume was weighed weekly and stored at −20 °C for the subsequent ammonia N (N-NH_3_) analysis.

Ruminal fluid was collected from goats after they left the chambers at the end of the experimental period. Ruminal fluid was taken just before the morning meal and 5 h after using an esophageal probe. To avoid saliva contamination, the first collected rumen sample (with the possible presence of saliva) was discarded. Rumen fluid samples were analyzed for pH, N-NH_3_, and VFA. The pH was measured immediately after sampling, whereas N-NH_3_ and VFA analyses were performed on samples stored at −20 °C.

Immediately after milking, the individual milk was weighed and a sample of 10% was placed in a bottle with an amount of about 10–15 mg of potassium dichromate per 100 g of milk as a preservative and stored at +4 °C.

### 2.3. Chemical Analyses

The linseeds and hemp seeds, TMR, orts, and feces were analyzed for chemical composition. The DM was determined by oven drying at 55 °C until a constant weight was obtained. Analytical DM was determined by drying in a ventilated oven at 100 °C overnight (AOAC International, 1995; method 945.15) [23]. The ash content was determined by incineration at 550 °C overnight in a muffle furnace (AOAC International, 1995; method 942.05) [23]. The CP was determined according to the Kjeldahl method (AOAC International, 1995; method 984.13) [23] using the Kjeltec Auto 1030 Analyzer (Tecator, Foss Analytical A/S, Hillerod, Denmark). The concentration of fiber was determined as described by Mertens [24], with the inclusion of heat-stable α-amylase and sodium sulfite, and expressed exclusive of residual insoluble ash (aNDFom). Acid detergent fiber and ADL, determined according to the method of Van Soest et al. [25], were expressed exclusive of residual insoluble ash (ADFom and ADLom, respectively); lignin was determined by solubilization of cellulose with sulfuric acid. The NDF and ADF analyses were performed using an Ankom 200 fiber analyzer (Ankom Technology Corp., Fairport, NY, USA). The ether extract was determined according to AOAC International, method 920.29 [23]. The gross energy of the TMR, orts, feces, urine, and milk was determined using an adiabatic calorimeter (IKA 4000; IKA Werke GmbH and Co. KG, Staufen, Germany). Urine and milk samples were placed in polyethylene bags, freeze-dried, and then burnt in the calorimeter [26]. The concentration of N in the acidified urine, in the condensed water collected in the chamber, in the fresh feces, and in composite milk samples was determined by the Kjeldahl method (AOAC International, 1995; method 984.13) [23] using the Kjeltec Auto 1030 Analyzer (Tecator, Foss Analytical A/S, Hillerød, Denmark).

Protein, casein, and NPN contents of milk were determined as described by Bava et al. [27]. The milk fat and lactose concentrations were determined using a Fourier transform infrared analyzer (MilkoScan FT6000; Foss Analytical A/S, Hillerod, Denmark). The milk urea level (MUL) concentration was determined using a differential pH technique (ISO, 2004; method 14637) [28]. The fat and protein corrected milk (FPCM) was referred to a standardized milk with 3.5, 3.2, and 4.5% of fat, crude protein, and lactose, respectively. The equation used to calculate FPCM was: FPCM = milk yield × (0.26 + 0.1352 × Fat% + 0.079 × CP%). This equation was obtained applying the following heats of combustion values for fat, true protein, non-protein nitrogen (NPN), and lactose: 9.29, 5.71, 2.21, and 3.95 Mcal/kg, respectively [29]. NPN was assumed to be 8.5% of milk CP (N × 6.38). Fatty acid composition was determined by gas chromatography (GC) after base-catalyzed transesterification of fat as described in Revello Chion et al. [30]. The SCC was performed with a Fossomatic 360 (Foss Analytical A/S, Hillerød, Denmark) and expressed as a linear score (LS = log_2_ (SCC/12,500)).

Rumen samples were analyzed for pH, N-NH_3_ with the Kjeltec Auto 1030 Analyzer (Tecator, Foss Analytical A/S, Hillerød, Denmark), and VFA using an Agilent 3000A micro GC gas chromatograph (Agilent Technologies, Santa Clara, CA, USA) according to Pirondini et al. [31].

### 2.4. Energy Metabolism Assumptions

Milk yield energy (kJ/BW^0.75^) was also corrected in function of the retained energy (kJ/BW^0.75^) as follows: corrected milk yield energy (MYEc) = milk yield energy + (1.014 × positive retained energy) [32] or MYEc = milk yield energy + (0.84 × negative retained energy) [33]. The efficiency of use of the metabolizable energy for lactation (k_l_) was calculated with the following equation: k_l_ = MYEc/(MEI − ME_m_), where MEI (kJ/BW^0.75^) is the metabolizable energy intake and ME_m_ (kJ/BW^0.75^) is the metabolizable energy for maintenance. The ME_m_ was determined by means of regression analysis between milk yield energy plus retained energy versus MEI for each treatment. Net energy for lactation (NE_l_) of the diet, expressed in MJ/kg DM, was computed as: ME × k_l_, where ME is the metabolizable energy concentration of the diet (MJ/kg DM); the milk fodder units (UFL) were calculated as: NE_l_/7.1128.

### 2.5. Statistical Analysis

The data obtained in the experiment were analyzed by the following proc. Mixed in SAS (SAS version 9.4, SAS Institute, Cary, NC, USA) [34]:$$Y_{ij}=µ+T_{i}+A_{j}\left(T\right)+e_{ij}$$
where *Y* is the dependent variable, µ is the overall mean, *T_i_* is the treatment effect (*i* = 1, 3), *A_j_*(*T*) is the random animal effect within treatment (*j* = 1, 4), *e_ij_* is the residual error.

Least squares mean estimates are reported. For all statistical analysis, significance was declared at *p* < 0.05 and trend at *p* < 0.1.

## 3. Results

### 3.1. Chemical Analysis of the Seeds and the Diets

Chemical analysis of linseeds, hemp seeds, and of the three diets, are reported in Table 2.

Considering the two whole seeds, hemp showed higher CP and aNDFom contents (27.3 and 34.0% on DM, respectively) compared to linseed (25.3 and 21.4% on DM). By contrast, EE content was higher for linseed than hemp (43.0 vs. 35.7% on DM), and this determined the slightly higher EE concentration for the L than the H diet. Hemp seeds are rich in linoleic acid (LA) (55.7 g/100 g fatty acids), whereas linseeds are rich in α-linolenic acid (ALA) (55.7 g/100 g fatty acids). The different fatty acid composition of seeds determined a high LA content for H (43.5 g/100 g FA) and a high ALA content for L (37.3 g/100 g FA).

Diets supplemented with oilseeds were characterized by higher EE (2.24, 5.96, and 5.12% on DM, respectively, for C, L, and H) and lower non-fiber carbohydrates (NFC) (37.2, 32.7, and 31.9% on DM, respectively, for C, L, and H) concentrations than C.

### 3.2. Dry Matter Intake and Apparent Diet Digestibility

The average dry matter intake was 1.91 kg/d, without differences among diets (Table 3).

The dry matter (64.4, 58.8, and 61.1 for C, L, and H, respectively; *p* = 0.04) and OM digestibilities (65.9, 60.0, and 62.5 for C, L, and H, respectively; *p* = 0.03) were lower for L than C, with H being intermediate. Ether extract digestibility was significantly higher (*p* = 0.02) for oil seed supplemented diets. The CP and the aNDFom digestibilities were on average 56.1 and 42.0%, respectively, without differences between diets. Energy digestibility of the L diet (59.3%) tended to be lower than those of H and C (62.2 and 64.7%, respectively; *p* = 0.09).

### 3.3. Ruminal Fermentation

The rumen fluid pH was on average 6.69 before the morning feeding and decreased to 6.55 after 5 h, without differences between treatments (Table 4).

The total VFA concentration was on average 125 mmol/L in the first sampling time and increased to 142 mmol/L after 5 h, with no differences between treatments. The VFA proportions were not affected by treatments and the acetate:propionate ratio was on average 4.1 at 0 h and 3.8 after 5 h.

Finally, ammonia nitrogen concentration was on average 11.5 and 6.8 mmol/L at 0 and 5 h, respectively, with no difference between treatments.

### 3.4. Milk Production and Fatty Acid Profile

The milk production and composition data are presented in Table 5.

Milk production was on average 2.30 kg/d without differences among diets. FPCM was not affected by the diet, nor was the milk composition in terms of CP and lactose. Milk fat percentage was influenced by dietary lipid supplementation (*p* = 0.016): goats fed H and L diets had a higher milk fat content (on average 4.07%) than C (3.20%). The dairy efficiency (kg FPCM/kg DMI) was on average 1.21 and was not affected by treatments. No differences between treatments were observed for MUL.

The milk fatty acid composition, expressed as g/100 g of total FA, is reported in Table 6. The sum of saturated linear chain fatty acids (SFA) was higher for C, followed by H and finally by L (70.9, 63.9, and 58.7%, respectively; *p* < 0.001). Linseed and hemp diet reduced C:16:0, C:14:0, and C:10:0 fatty acids compared to C (*p* < 0.001). The amount of monounsaturated fatty acids (MUFA) was higher for L, intermediate for H, and lower for C diet (33.3, 29.6, and 22.4%, respectively; *p* < 0.001). Oleic acid was increased by L and H (*p* = 0.003). The sum of polyunsaturated fatty acids (PUFA) was increased only by L, whereas no difference was seen between H and C (4.79, 3.65, and 3.52%, respectively; *p* = 0.004). ALA, as expected, was highest in L (*p* < 0.001), due to its high content in linseeds. On the contrary, no differences were registered for LA among diets, despite the high LA content of hemp. The n-6/n-3 ratio was statistically influenced by treatments, with L having the lowest value (1.21) and C the highest (5.43) (*p* < 0.001). The amount of preformed FA (>C:16:0) was significantly increased by L, followed by H, whereas C recorded a lower value (57.5, 50.1, and 34.7% respectively; *p* < 0.001).

### 3.5. Gas Exchange and Methane Production

The oxygen consumption, CO_2_, and CH_4_ production are reported in Table 7.

No difference in O_2_ consumption and CO_2_ production was observed between treatments. This was also confirmed by the similar values of the three respiratory quotients. The dietary treatments did not affect methane emission, either in terms of CH_4_ production or that related to different dietary variables or milk production.

### 3.6. Nitrogen and Energy Balance

The results concerning the N balance are presented in Table 8.

The N intake was 37.7, 38.4, and 45.6 g/d for C, L, and H, respectively. No differences were observed between treatments for fecal and urinary nitrogen excretions, both in absolute values and as a percentage of N intake. In addition, the values of milk N, expressed in g/d, were also highly similar, whereas H showed a significantly lower efficiency of dietary nitrogen utilization for milk production (milk N as % of N intake; *p* = 0.014). Finally, a tendency (*p* = 0.082) was found regarding the N balance (N retained as % of N intake): the goats fed H had a positive N balance (+5.44%) compared to those fed C or L (−4.69 and −1.89%, respectively).

The results concerning the energy balance are reported in Table 9.

The ME_m_, computed by regression analysis between milk yield energy plus retained energy versus MEI, was found to be 482 (kJ/BW^0.75^).

The individual energy intake was on average 37.4 MJ/d without differences between treatments. No differences were observed for digestible energy and the energy lost with feces, urine and methane. Moreover, the diets did not affect the metabolizable energy, heat production, milk energy, or the retained energy.

## 4. Discussion

### 4.1. Chemical Composition and Digestibility

In the present trial, the inclusion in the diet of almost 10% of linseeds or hemp seeds was evaluated.

Studies conducted about the use of hemp in ruminant diets have mainly evaluated hemp cake [19,20] rather than seeds. To the best of our knowledge, only Gibb et al. [37] conducted an in vivo experiment testing two additional levels of hemp seeds (9 and 14%) in feedlot cattle diets, with the lower level similar to that used in the present study.

More studies were conducted regarding the use of linseed, mostly in lactating cows [7,38,39,40], whereas few studies [3,41,42] evaluated linseed in goat diets with variable inclusion levels. In particular, Bernard et al. [41] used an amount of extruded linseed of about 22% on DM, which is a much higher level than that tested in the present study.

The chemical analysis of hemp seeds returned values in line with those reported by other authors [10,16,43]. However, it must be underlined that hemp seeds have a high variable chemical composition depending on their origin and variety [44].

Linseed composition, expressed on a DM basis, was similar to that reported by Wang et al. [10], with the exception of a higher EE (43 vs. 38%) and a lower aNDFom (21 vs. 31%) content. Comparison of the fatty acid composition of linseeds and hemp seeds shows an inversion of the relationship between linoleic acid and α-linolenic acid: linseeds are richer in ALA, whereas hemp seeds have more LA. The 3.47 ratio of LA to ALA found in hemp seeds is relatively close to the 3:1 ratio which is assumed to be optimal for human nutrition [11].

In cows, lipid supplementation is often associated with a decrease in dry matter intake, which was not observed in the present study on goats. The results of Benchaar et al. [45] on cows showed that at rates ranging from 3.7 to 5.5% of total DM, linseed oil supplementation reduced intake and DM digestibility in high-starch diets, but not in high-fiber diets such as the diet of the present study (on average 40.3% of aNDFom on DM).

The inclusion of linseeds reduced both DM and OM digestibilities by about 9% compared to C. Similar results were reported by Chung et al. [46], who found a 7% reduction in DM and OM digestibilities in diets with the inclusion of approximately 9% ground linseed in non-lactating cows. Similarly, again for dairy cows, Martin et al. [38] showed that supplying 5.7% lipids from linseed significantly reduced OM and NDF digestibility. Other authors observed no effects of linseed oil supplementation (4% of DM) on DM intake and digestibility when the forage portion of the diet consisted mainly or completely of non-cereal forages [47,48]. The decrease in OM digestibility of diet L in comparison with C is higher than that calculated by taking into account the OM digestibility coefficients [49] of linseed, soybean meal, and corn flakes.

Regarding hemp seeds, to the best of our knowledge, no in vivo studies have been conducted to determine the effects of the inclusion of hemp seeds on diet digestibility in ruminants and the results of the present study indicate that hemp seeds, in contrast to linseeds, can partially substitute soybean meal in the ration of lactating goats without any adverse effects on diet digestibility.

The supplementation of both linseeds and hemp seeds increased the EE digestibility by about 22% compared to C. This is consistent with the study of Benchaar et al. [45] on rumen cannulated lactating cows fed two diets based on red clover silage or corn silage supplemented with linseed oil, where the authors found an increase in EE digestibility of about 15%.

### 4.2. Rumen Fermentation Profile

Rumen fermentation profile was not affected by diet.

The rumen pH values were registered just before the morning feeding and 5 h after; the nadir pH appeared after this time interval and allowed the differences in diets and possible sub-acidosis issues to be better understood. However, no differences of pH were observed between diets for both sampling times. Although not significant, the pH of C had the greater decrease between the two sampling times, probably due to the higher NFC content and the subsequent VFA production in comparison with the other two diets.

The acetate:propionate ratio was not affected by the treatment, contrary to the observation of Chung et al. [46], who found a reduction in the ratio with the inclusion of linseeds. The acetate:propionate ratio was found to be higher than that normally found in cattle, and consistent with that reported for goats by Nozière et al. [50].

Benchaar et al. [45] did not observe any effects on rumen fermentation parameters in lactating cows fed red clover diets supplemented with linseed oil. However, the same authors observed a decline in total VFA concentration with linseed oil supplementation to a corn silage-based diet. This decline was consistent with a decrease in fiber digestibility observed for the corn silage-based diet.

As expected, ammonia N decreased 5 h after feeding, probably due to the microbial growth in this phase, but no difference was registered between treatments.

### 4.3. Milk Yield and Fatty Acid Composition

Milk yield was not affected by oilseed supplementation, in accordance with the results of Chilliard et al. [4] in goats fed diets with lipid supplementation. This is consistent with the findings reported by Mele et al. [51], who noted that, in goats, the effect of fat addition to the diet in mid lactation increases the milk fat content without an increase in milk yield. Contrary to the findings observed in this study, Meignan et al. [52] reported that extruded linseed supplementation increased milk yield (0.72 kg/d) in a meta-analysis on cows. Similarly, Benchaar et al. [48] reported a linear increase in cow milk yield with the increasing inclusion of linseed oil. In a study on the potential for using hemp cake as feed for dairy cows, Karlsson et al. [19] observed significant quadratic effects of the inclusion of hemp cake on milk and energy-corrected milk yields. In another trial, Mierliţă [20] compared a control diet (composed of hay, high-energy concentrates, and sunflower meal) with two experimental diets that were formulated to provide the same amount of fat via hemp seed or hempseed cake: Both treatments significantly increased the yields of milk and energy-corrected milk.

In our study, the treatments had no effects on the FPCM, CP, lactose, and fat yield; however, the fat percentage was significantly increased by linseeds and hemp seeds. In cows, lipid supplementation is often associated with fat milk depression, whereas in goats, lipid supplementation generally increases milk fat [4], as confirmed by the present study results. For example, Karlsson et al. [19] showed a linear decrease in milk fat content of cows fed diets characterized by a different inclusion of hemp cake (from 0 to 32% on DM of hemp cake with a ration fat content from 2.3 to 5.4% on DM), whereas Mierliţă [20] reported a significant increase in milk fat content and fat milk yield in sheep fed hemp seed or hemp seed cake.

Lipid supplementation in dairy ruminants is a well-known strategy to modify the milk FA composition [53], as confirmed by the data of the present study. As expected, L and H diets caused a reduction in the synthesis of de novo FA. In particular, the decrease in de novo FA was greater for L than for H (−35.9 and −23.9%, respectively, compared to C). The reduction mainly concerned C10:0, C12:0, C14:0, and C16:0, whereas among short-chain FA (C < 8), the only reduction concerned C6:0, and only for the L diet. As reported by Chilliard et al. [36], in goat milk, physiological variations in the percentages of C4:0 to C8:0 FA are not well related to those of C10:0 to C16:0 FA, as emerged in this work. Moreover, the reduction of de novo FA can be attributed to the availability of long-chain FA in the diet, which are potent inhibitors of mammary FA synthesis, through a direct inhibitory effect on acetyl-CoA carboxylase activity, as reported by Chilliard and colleagues [36]. On the contrary, long-chain FA are not synthesized by ruminant cells; therefore, their concentration in milk is closely related to the amount of long-chain fatty acids ingested with the diet [53]. In our study, the percentage of preformed FA (>16:0) was significantly increased by L and H diets. Cremonesi et al. [54] found higher amounts of C18:0 (24.6, 43.4, and 35.2% of total FA, for C, L, and H, respectively) in rumen fluid of goats fed the same diets used in the present experiment, confirming the extensive biohydrogenation occurring in the rumen.

The amount of C18:0 found in milk reflects the pattern seen in the rumen; moreover, a higher amount of C18:1 *cis*-9 was found for L and H diets. For the oilseed diets, the trans-MUFA also increased significantly, proportional to the MUFA and PUFA contents of the diets, confirming the findings in rumen fluid of Cremonesi et al. [54].

The linseed diet significantly increased the concentrations C18:3 n-3, the most abundant FA in linseeds, in line with earlier studies in lactating goats, ewes, and cows [7,41,55]. Although the hemp diet, which is rich in C18:2 n-6, showed a higher concentration of this FA in rumen fluid [53], no effect was detected in the milk.

Both L and H diets, compared to C, reduced the sum of SFA, mostly due to the reduction of short-chain SFA, in line with earlier studies [7,41,55] and increased MUFA, whereas only the L diet increased PUFA, particularly ALA, related to its high content in the diet. This, in turn, decreased the milk n-6/n-3 ratio of oilseed treatments, particularly L, in comparison with diet C, as was observed by Bernard et al. [41].

### 4.4. Methane Emission

Lipid supplementation is a strategy used to lower enteric CH_4_ emission by ruminants. In this regard, the addition of fat to the diet reduces ruminal OM degradability, decreases DMI, lowers the number of protozoa and, therefore, of associated methanogens, and has a direct toxic effect on methanogens [5,45,56].

In our study, no differences were seen for gas exchange from the goats fed the experimental diets. The recorded values of methane production of our study are in line with those calculated using the equations for small ruminants proposed by Tamburini et al. [57]. Moreover, the methane conversion factors (*Y*_m_; *Y*_m_ = methane energy/GE intake) of this study are very close to those calculated with the formula proposed by the FAO [58]. No treatment effect was recorded for all of the values of methane production, not only in absolute terms but also in relation with other variables (e.g., as % of GE intake). In an in vitro study, Wang et al. [10] added 35 and 70 g kg^−1^ of both hemp seeds and linseeds (in the two forms, whole and extruded). They found a significant reduction of total gas production and in vitro organic matter degradability at both dosages in hemp and linseed diets compared to the control diet. When added at the lowest dosage, only hemp seeds reduced CH_4_ per unit of DM (−9.8%), whereas both oilseeds were able to reduce CH_4_ per unit of DM when added at the highest dosage, although hemp seeds achieved the greatest effect (−17 and −9.5%, respectively, for hemp seeds and whole linseeds).

In an in vivo study, Poteko et al. [40] evaluated the effects of adding extruded linseeds (29 g kg^−1^ DM) to a control diet on CH_4_ emission of cows using a tracer gas technique and in respiration chambers. For both methods, they did not find a reduction in CH_4_ (g/d; g/kg of DMI; g/kg of OM intake and g/kg of energy corrected milk). In another study, Martin et al. [38] evaluated the effects of adding crude linseed, extruded linseed, or linseed oil to a control diet on methane production in dairy cows. In contrast to Poteko et al. [40], Martin et al. [38] found that all treatment diets significantly reduced daily CH_4_ emissions, but to different extents (−12% with crude linseed, −38% with extruded linseed, −64% with linseed oil) compared with the control diet. The same ranking among diets was observed for CH_4_ as a percentage of energy intake or grams of CH_4_ per kilograms of OM intake. However, they did not find any significant difference in terms of CH_4_ production per kg of digested OM between control and crude linseed diets, in accordance with our study, underlying the importance of the physical form of the seeds.

The lack of the effect of whole oilseed diets on methane emissions can probably be ascribed to the physical form of the seeds, which favored at least a partial rumen bypass of their lipids, in agreement with the findings of a previous work [26], in which the inclusion (4.7% of DM) of rumen inert fat (calcium soaps of palm oil FA) in a goat diet did not reduce methane emissions. In addition, the physical form of the seeds would have prevented the lipid supplementation from reducing the archaeal abundance in rumen fluid, as seen in the work of Cremonesi et al. [54].

### 4.5. Nitrogen Balance

Regarding nitrogen balance, the mean values obtained for both fecal and urinary excretion are consistent with those obtained in previous studies [27,59]. The relatively low N digestibility of the three diets can mainly be ascribed to the poor quality of the forages utilized. Although not significant, the lower N fecal excretion with diet C can be attributed to its higher content in soybean meal, which is a raw material characterized by higher CP digestibility (80%; 60]) that that of whole linseed (73%; [60]) and hemp seeds (75%; [61]).

The higher N fecal excretion with the high lipid diets were counterbalanced by a lower N urinary excretion. Hence, the total N manure excretion was similar for the three diets.

The goats fed the H diet, having numerically higher DMI (+17%) in comparison with the other two diets but the same milk yield, showed a trend (*p* = 0.082) for a higher N retention (% of N intake). This is in line with the higher value of retained energy for the H diet, although not significant.

### 4.6. Energy Balance

Considering the energy balance, the value found for ME_m_ (482 kJ/BW^0.75^) is very similar to that (484 kJ/BW^0.75^) previously found by our group on lactating multiparous Saanen goats [32].

The trend for a lower digestible energy for the L diet is linked to the lower OM and NDF digestibility, and this is consistent with the findings of Martin et al. [62] in a study on dairy cows.

In general, the results obtained in the present trial did not reveal evidence of effects of whole linseeds and hemp seeds on the energy utilization of the diet, other than a trend for a lower DE and ME of the two treatment diets in comparison with the control. These trends reflect the significant differences in OM and EE digestibility between C and L diets. The higher EE digestibility of the L diet, due to the higher calorific value of lipids in comparison with the other nutrients, partially compensate for its lower OM digestibility, resulting in a trend of lower GE digestibility of this diet compared to the control.

The numerically lower DE of the L diet results in a lower ME of the same diet. The values of methane energy losses, similar among the diets, are in line with those obtained in a previous experiment [59], and consistent with those obtained in dry goats by Lima et al. [63] as the ratio between methane energy loss and ME intake.

The average k_l_ found in this study (0.685) is consistent with literature values for goats of 0.67 [64,65,66].

## 5. Conclusions

The objective of this study was to evaluate the effect of including whole linseeds and hemp seeds in the diet of dairy goats to increase milk fat content, modify milk fatty acid composition, and reduce methane production.

The results of the present work suggest that supplementing whole oilseeds can be an effective strategy to increase milk fat concentration and modify milk fatty acid composition, with advantages in terms of milk quality, which is also linked to benefits for human health. Regarding methane emissions, however, the results of this study indicate that neither linseed nor hemp seeds significantly reduce methane production.

## Figures and Tables

**Table 1 animals-11-02717-t001:** Composition of the basal ration (85.6% of total diet DM) and of the three different concentrate supplements (14.4% of total diet DM) used in the experiment (% of DM).

Composition	Diet ^1^		
	C	L	H
Basal ration			
Meadow hay, 2nd cut	28.4	28.4	28.4
Meadow hay, 1st cut	16.3	16.3	16.3
Alfalfa hay, 3rd cut	12.2	12.2	12.2
Corn flakes	15.3	15.3	15.3
Soybean hulls	12.8	12.8	12.8
Corn grain, meal	6.37	6.37	6.37
Carob pulp	5.23	5.23	5.23
Soybean meal	3.42	3.42	3.42
Concentrate supplements			
Linseed, whole	-	65.2	-
Hemp seed, whole	-	-	65.3
Corn flakes	38.7	-	-
Soybean meal	26.0	-	-
Soybean hulls	19.9	19.6	19.6
Sugar cane, molasses	2.80	2.76	2.75
Salts ^2^	10.1	10.0	10.0
Amino acid and vitamin supplement ^3^	1.48	1.46	1.45
Vitamin–mineral mix ^4^	1.01	1.00	1.00

^1^ C: control diet; L: linseed diet; H: hemp seed diet. ^2^ Salts: 34% calcium carbonate, 31.9% sodium bicarbonate, 14.9% dicalcium phosphate, 12.8% sodium chloride, 6.4% magnesium oxide. ^3^ Rumen-protected amino acid and vitamin supplement provided (per kg): 200 g D-L methionine, 162 g of choline chloride, 25 g of betaine, 1 g of vitamin B_2_. ^4^ Provided (per kg): 3500 kIU of vitamin A, 350 kIU of vitamin D_3_, 7000 mg of vitamin E, 1000 mg of vitamin B_1_, 1000 mg of vitamin B_2_, 1000 mg of vitamin B_6_, 1.5 mg of vitamin B_12_, 8000 mg of nicotinic acid, 215 mg of vitamin K, 40 mg of biotin, 24,000 mg of choline chloride, 6370 mg of ferrous carbonate, 6840 mg of ferrous sulfate monohydrate, 327 mg of potassium iodide, 9820 mg of cupric sulfate pentahydrate, 6750 mg of manganese oxide, 20,690 mg of manganese sulfate monohydrate, 10,920 mg of manganese chelate of amino acids hydrate, 6500 mg of zinc oxide, 14,400 mg of zinc sulfate monohydrate, 22,900 mg of zinc chelate of amino acids hydrate, 87.6 mg of sodium selenite.

**Table 2 animals-11-02717-t002:** Chemical analysis (% of DM, unless otherwise stated) of linseeds, hemp seeds, and the three experimental diets.

Item	Whole Seeds		Diet ^1^		
	Lin	Hemp	C	L	H
Chemical composition					
DM, %	93.6	93.7	87.9	88.0	88.0
OM	97.1	94.8	92.1	91.9	92.2
CP	25.3	27.3	13.4	13.4	13.7
EE ^2^	43.0	35.7	2.24	5.96	5.12
aNDFom ^3^	21.4	34.0	39.3	39.9	41.6
ADFom ^4^	13.6	28.2	30.1	30.8	32.4
ADLom ^5^	8.2	12.4	6.04	6.59	6.97
NFC ^6^	7.47	-	37.1	32.7	31.9
GE ^7^, MJ/kg DM	27.1	26.6	18.76	19.47	19.43
Fatty acid composition (g/100 g FA)					
16:0, palmitic acid	5.21	7.16	28.01	13.14	17.50
18:0, stearic acid	3.87	4.44	16.89	7.34	8.64
18:1 *cis*-9 (n-9), oleic acid	17.40	12.94	18.25	21.58	18.77
18:2 *cis-9*, *12* (n-6), linoleic acid	16.25	55.73	28.54	20.41	43.46
18:3 *cis*-6, *9*, *12* (n-6), GLA ^8^	0.43	2.32	0.06	0.14	1.06
18:3 *cis-9, 12, 15* (n-3), ALA ^9^	55.70	16.10	8.05	37.30	10.28
18:4 *cis-6, 9, 12, 15* (n-3), SDA ^10^	0.00	0.64	0.05	0.04	0.27
Other fatty acids	1.15	0.67	0.15	0.05	0.02
(n-6)/(n-3)	0.30	3.47	3.53	0.55	4.22

^1^ C: control diet; L: linseed diet; H: hemp seed diet. ^2^ Ether extract. ^3^ aNDFom: neutral detergent fiber determined with the addition of α-amylase and sodium sulfite and corrected for insoluble ash. ^4^ ADFom: acid detergent fiber corrected for insoluble ash. ^5^ ADLom: acid detergent lignin corrected for insoluble ash. ^6^ NFC: non-fiber carbohydrates calculated as follows: NFC = 100 − (ASH + CP + EE + aNDFom). ^7^ GE: gross energy. ^8^ GLA: γ-linolenic acid. ^9^ ALA: α-linolenic acid. ^10^ SDA: stearidonic acid.

**Table 3 animals-11-02717-t003:** Dry matter intake and apparent diet digestibility (%).

Item	Diet ^1^			SEM	*p*-Value
	C	L	H		
DMI, kg/d	1.81	1.81	2.12	0.28	0.438
DM	64.4 ^a^	58.8 ^b^	61.1 ^a,b^	1.37	0.039
OM	65.9 ^a^	60.0 ^b^	62.5 ^a,b^	1.37	0.030
ASH	46.6	45.3	43.0	1.89	0.337
EE ^2^	47.5 ^b^	57.7 ^a^	58.6 ^a^	2.60	0.016
CP	58.6	54.5	55.4	2.00	0.297
aNDFom ^3^	44.6	39.1	42.3	2.51	0.303
ADFom ^4^	44.7	37.8	42.3	3.27	0.324
NFC ^5^	92.9	89.6	92.0	1.12	0.128
GE ^6^	64.7	59.3	62.2	1.59	0.089

^1^ C: control diet; L: linseed diet; H: hemp seed diet. ^2^ Ether extract. ^3^ aNDFom: neutral detergent fiber determined with the addition of α-amylase and sodium sulfite and corrected for insoluble ash. ^4^ ADFom: acid detergent fiber corrected for insoluble ash. ^5^ NFC: non-fiber carbohydrates calculated as follows: NFC = 100 − (ASH + CP + EE + aNDFom). ^6^ GE: gross energy. ^a,b,c^: different letters on the same row correspond to different Least Square Means (*p* < 0.05).

**Table 4 animals-11-02717-t004:** Rumen fermentation parameters of the goats fed the experimental diets, sampling the rumen fluid just before the morning meal and 5 h after.

Item	Hours ^2^	Diet ^1^			SEM	*p*-Value
		C	L	H		
pH	0	6.84	6.69	6.53	0.27	0.665
	5	6.54	6.66	6.45	0.18	0.695
Total VFA, mmol/L	0	130	112	132	14.1	0.550
	5	167	123	136	39.9	0.688
VFA, mol/100 mol VFA						
Acetate	0	70.9	73.4	72.9	1.00	0.172
	5	72.2	74.6	71.1	0.93	0.053
Propionate	0	18.5	18.8	16.8	1.41	0.499
	5	19.4	16.8	18.8	0.89	0.141
Isobutyrate	0	1.21	1.00	0.91	0.28	0.682
	5	0.46	0.44	0.59	0.38	0.945
Butyrate	0	8.61	6.33	8.56	1.09	0.270
	5	7.47	7.49	9.09	1.12	0.446
Isovalerate	0	0.36	0.19	0.33	0.16	0.699
	5	0.04	0.14	0.05	0.03	0.142
n-valerate	0	0.49	0.31	0.54	0.11	0.311
	5	0.40	0.45	0.40	0.07	0.869
Acetate:propionate	0	3.90	3.93	4.44	0.36	0.431
	5	3.73	4.45	3.16	0.37	0.86
Ammonia N, mmol/L	0	11.4	9.19	13.9	1.77	0.203
	5	6.13	6.80	7.61	1.32	0.668

^1^ C = control diet; L = linseed diet; H = hemp seed diet. ^2^ Hour after morning meal.

**Table 5 animals-11-02717-t005:** Milk yield and composition of the goats fed the three experimental diets.

Item	Diet ^1^			SEM	*p*-Value
	C	L	H		
Milk yield, kg/d	2.31	2.23	2.35	0.193	0.897
FPCM ^2^, kg/d	2.18	2.31	2.51	0.195	0.412
Dairy efficiency ^3^	1.18	1.29	1.17	0.076	0.479
Fat, %	3.20 ^b^	3.94 ^a^	4.21 ^a^	0.213	0.016
Fat yield, g/d	75.1	89.4	98.8	9.00	0.166
Crude protein, %	3.11	3.05	3.01	0.050	0.285
True protein, %	2.93	2.75	2.80	0.108	0.529
Protein yield, g/d	71.8	67.5	70.6	5.55	0.840
Lactose, %	4.44	4.40	4.32	0.111	0.649
Non-protein N, % total N	8.61	9.03	7.86	1.06	0.703
Casein N, % total N	74.0	73.1	74.9	2.35	0.858
MUL ^4^, mg/dL	30.2	29.1	29.2	3.60	0.961
Linear score ^5^	5.06	6.48	4.52	0.782	0.217

^1^ C: control diet; L: linseed diet; H: hemp seed diet. ^2^ FPCM: fat and protein corrected milk = milk yield × (0.26 + 0.1352 × Fat% + 0.079 × CP%). ^3^ Dairy efficiency: FPCM/DMI. ^4^ MUL: milk urea level ^5^ Linear score: log_2_ (somatic cell count/12,500). ^a,b^: Different letters in the same row correspond to statistically different values (*p* < 0.05).

**Table 6 animals-11-02717-t006:** Milk fatty acid (FA) composition (g/100 g of total FA) as affected by the three experimental diets.

Fatty Acid		Diet ^1^		SEM	*p-*Value
	C	L	H		
4:0, butyric acid	2.13	2.39	2.44	0.103	0.081
6:0, caproic acid	2.44 ^a^	2.12 ^b^	2.46 ^a^	0.083	0.030
7:0, enanthic acid	0.036	0.038	0.046	0.015	0.841
8:0, caprylic acid	2.78 ^a^	2.02 ^b^	2.55 ^a^	0.124	0.005
9:0, pelargonic acid	0.066	0.040	0.048	0.007	0.055
10:0, capric acid	10.4 ^a^	6.00 ^c^	7.92 ^b^	0.321	<0.001
10:1, caproleic acid	0.236 ^a^	0.119 ^c^	0.170 ^b^	0.014	<0.001
11:0, undecanoic acid	0.117 ^a^	0.058 ^b^	0.067 ^b^	0.013	0.015
12:0, lauric acid	5.30 ^a^	2.58 ^b^	3.19 ^b^	0.221	<0.001
12:1, lauroleic acid	0.048	0.017	0.018	0.014	0.182
*iso* 13:0, iso-tridecanoic acid	0.025	0.029	0.022	0.003	0.369
*anteiso* 13:0, ante iso-tridecanoic acid	0.050 ^a^	0.016 ^b^	0.027 ^b^	0.006	0.005
13:0, tridecanoic acid	0.164 ^a^	0.083 ^b^	0.100 ^b^	0.011	<0.001
*iso* 14:0, iso-myristic acid	0.139	0.121	0.130	0.008	0.302
14:0, myristic acid	11.2 ^a^	6.95 ^b^	8.14 ^b^	0.500	<0.001
14:1 *cis*-9 (n-5), myristoleic acid	0.174 ^a^	0.074 ^b^	0.106 ^b^	0.017	0.006
*iso* 15:0, iso-pentadecanoic acid	0.286 ^a^	0.249 ^b^	0.250 ^b^	0.010	0.035
*anteiso* 15:0, ante iso-pentadecanoic acid	0.412	0.354	0.350	0.020	0.064
15:0, pentadecanoic acid	1.042 ^a^	0.774 ^b^	0.816 ^b^	0.038	0.001
15:1, pentadecenoic acid	0.167	0.135	0.149	0.065	0.933
*iso* 16:0, iso-palmitic acid	0.187	0.160	0.166	0.051	0.909
16:0, palmitic acid	26.8 ^a^	16.8 ^c^	19.6 ^b^	0.740	<0.001
16:1 cis-9 (n-7), palmitoleic acid	0.815 ^a^	0.455 ^b^	0.627 ^b^	0.061	0.006
*iso* 17:0, iso-heptadecanoic acid	0.436	0.439	0.411	0.041	0.849
*anteiso* 17:0, ante iso-heptadecanoic acid	0.274	0.247	0.274	0.027	0.700
17:0, heptadecanoic acid	0.918 ^a^	0.638 ^b^	0.662 ^b^	0.039	<0.001
17:1 *cis*-9, cis-9 heptadecenoic acid	0.118	0.074	0.077	0.035	0.552
*iso* 18:0, iso-stearic acid	0.170	0.093	0.117	0.047	0.475
18:0, stearic acid	7.56 ^c^	18.1 ^a^	15.6 ^b^	0.790	<0.001
18:1 *trans*-6/7/8, trans-6,7,8-octadecenoic acid	0.077 ^b^	0.193 ^a^	0.223 ^a^	0.025	0.003
18:1 *trans*-9, elaidic acid	0.192 ^b^	0.313 ^a^	0.322 ^a^	0.022	0.003
18:1 *trans*-10, trans-10-octadecenoic acid	0.211 ^b^	0.360 ^a^	0.500 ^a^	0.050	0.005
18:1 *trans*-11, trans-vaccenic acid	0.672 ^b^	0.989 ^a^	1.104 ^a^	0.076	0.005
18:1 *trans*-12, trans-12-octadecenoic acid	0.150 ^b^	0.620 ^a^	0.576 ^a^	0.050	<0.001
18:1 *trans*-13/14, trans-13/14-octadecenoic acid	0.233 ^b^	1.161 ^a^	0.690 ^ab^	0.192	0.019
18:1 *trans*-16, trans-16-octadecenoic acid	0.165 ^c^	1.12 ^a^	0.600 ^b^	0.047	<0.001
18:1 *cis*-9 (n-9), oleic acid	17.4 ^b^	23.8 ^a^	21.9 ^a^	1.014	0.003
18:1 *cis*-11, cis-vaccenic acid	0.634	0.506	0.488	0.047	0.074
18:1 *cis*-12, cis-12-octadecenoic acid	0.229 ^b^	0.526 ^a^	0.539 ^a^	0.039	<0.001
18:1 *cis*-13, cis-13-octadecenoic acid	0.053 ^c^	0.113 ^a^	0.074 ^b^	0.007	<0.001
18:1 *cis*-15, cis-15-octadecenoic acid	0.111 ^b^	0.747 ^a^	0.216 ^b^	0.065	<0.001
18:2 *cis*-9 *trans*-12, linoleadic acid	0.586 ^c^	1.98 ^a^	1.09 ^b^	0.105	<0.001
18:2 *cis*-9 *trans*-11 (CLA ^2^), rumenic acid	0.345	0.379	0.430	0.033	0.160
18:2 *cis*-9 *cis*-12 (n-6), linoleic acid	2.56	2.25	2.47	0.159	0.381
18:3 *cis*-9 *cis*-12 *cis*-15 (n-3), α-linolenic acid (ALA)	0.453 ^b^	1.91 ^a^	0.559 ^b^	0.074	<0.001
18:3 *cis*-6 *cis*-9 *cis*-12 (n-6), γ-linolenic acid (GLA)	0.033	0.019	0.056	0.014	0.189
20:0, arachidic acid	0.143 ^b^	0.186 ^b^	0.292 ^a^	0.016	<0.001
20:1 *cis*-11 (n-9), eicosenoic acid	0.007	0.011	0.006	0.003	0.421
20:2 *cis*-11,14 (n-6), eicosadienoic acid	0.017	0.055	0.021	0.016	0.224
20:3 *cis*-8,11,14 (n-6), dihomo-γ-linolenic acid	0.028	0.060	0.020	0.021	0.377
20:3 *cis*-11,14,17 (n-3), eicosatrienoic acid	0.006	0.003	0.021	0.012	0.504
20:4 *cis*-5,8,11,14 (n-6), arachidonic acid	0.007	0.002	0.002	0.002	0.091
20:5 *cis*-5,8,11,14,17 (n-3), EPA ^3^	0.022	0.039	0.021	0.010	0.370
22:0, behenic acid	0.004	0.005	0.005	0.001	0.586
22:1, erucic acid	0.085	0.046	0.052	0.028	0.535
22:5 *cis*-4,8,12,15,19 (n-3), cuplanodonic acid	0.004	0.010	0.002	0.004	0.408
22:6 *cis*-4,7,10,13,16,19 (n-3), DHA ^4^	0.014 ^b^	0.042 ^a^	0.015 ^b^	0.007	0.024
24:0, lignoceric acid	0.025	0.040	0.024	0.011	0.514
24:1, nervonic acid	0.003	0.004	0.004	0.002	0.653
Σ SFA ^5^	70.9 ^a^	58.7 ^c^	63.9 ^b^	1.05	<0.001
Σ MUFA ^6^	22.4 ^c^	33.3 ^a^	29.6 ^b^	1.06	<0.001
Σ *cis*-MUFA	20.1 ^b^	26.6 ^a^	24.5 ^a^	0.992	0.003
Σ *trans*-MUFA	2.29 ^c^	6.74 ^a^	5.11 ^b^	0.336	<0.001
Σ PUFA ^7^	3.52 ^b^	4.79 ^a^	3.65 ^b^	0.207	0.004
Σ PUFA (n-6)	2.67	2.41	2.60	0.166	0.496
Σ PUFA (n-3)	0.499 ^b^	2.00 ^a^	0.618 ^b^	0.081	<0.001
n-6/n-3 ratio	5.43 ^a^	1.21 ^c^	4.34 ^b^	0.37	<0.001
Σ BCFA ^8^	1.979	1.709	1.748	1.140	0.313
Σ *iso* BCFA	1.24	1.09	1.10	0.13	0.587
Σ *anteiso* BCFA	0.736 ^a^	0.617 ^b^	0.652 ^b^	0.030	0.037
OIAR ^9^	1.05	1.14	1.06	0.044	0.277
DI ^10^	0.015	0.010	0.013	0.001	0.091
<16:0 ^11^	64.1 ^a^	41.1 ^c^	48.8 ^b^	1.20	<0.001
>16:0 ^12^	34.7 ^c^	57.5 ^a^	50.1 ^b^	1.26	<0.001

^1^ C: control diet; L: linseed diet; H: hemp seed diet. ^2^ CLA: conjugated linoleic acids. ^3^ EPA: eicosapentaenoic acid. ^4^ DHA: docosahexaenoic acid. ^5^ SFA: saturated linear chain fatty acids. ^6^ MUFA: monounsaturated fatty acids. ^7^ PUFA: polyunsaturated fatty acids. ^8^ BCFA: branched-chain fatty acids. ^9^ OIAR: ratio of odd-*iso* to odd-*anteiso* FA: (*iso* 15:0 + *iso* 17:0)/(*anteiso* 15:0 + *anteiso* 17:0). ^10^ DI: desaturation index (*cis*-9 14:1/14:0 + *cis*-9 14:1). ^11^ <16:0: de novo fatty acids calculated according to Fievez et al. [35]. ^12^ >16:0: preformed FA calculated according to Chilliard et al. [36] and Fievez et al. [35]. ^a, b, c^: different letters in the same row correspond to statistically different values (*p* < 0.05).

**Table 7 animals-11-02717-t007:** Gas exchange and methane production of the goats fed the experimental diets.

Item	Diet ^1^			SEM	*p*-Value
	C	L	H		
O_2_ consumption, L/d	570	561	619	42.9	0.540
CO_2_ production, L/d	620	605	690	54.2	0.457
RQ ^2^	1.09	1.08	1.11	0.022	0.390
CH_4_ production, L/d	53.0	48.1	60.9	6.89	0.396
CH_4_, g/d	38.0	34.5	43.7	4.94	0.396
CH_4_, g/kg DMI	20.9	18.9	20.7	1.46	0.582
CH_4_, g/kg OM intake	31.7	28.8	31.2	2.21	0.600
CH_4_, g/kg dOM ^3^	47.9	48.1	49.8	3.14	0.863
CH_4_, g/kg NDF intake	52.5	46.7	50.2	4.05	0.579
CH_4_, g/kg dNDF ^4^	117	121	119	11.6	0.971
CH_4_, % GE intake	6.07	5.27	5.78	0.43	0.415
CH_4_, % DE intake	9.35	8.29	9.32	0.68	0.882
CH_4_, g/kg milk	16.4	14.8	19.6	1.53	0.103
CH_4_, g/kg FPCM ^5^	17.6	14.7	18.1	1.98	0.452

^1^ C = control diet; L = linseed diet; H = hemp seed diet. ^2^ RQ = respiratory quotient (CO_2_ production/O_2_ consumption). ^3^ dOM: digested OM. ^4^ dNDF: digested NDF. ^5^ FPCM: fat and protein corrected milk.

**Table 8 animals-11-02717-t008:** Nitrogen balance of the goats fed the experimental diets.

Item	Diet ^1^			SEM	*p*-Value
	C	L	H		
Feces DM, kg/d	0.645	0.749	0.828	0.086	0.278
Urine, kg/d	2.05	2.11	2.08	0.290	0.998
N intake, g/d	37.7	38.4	45.6	5.73	0.322
N excretion, g/d					
Fecal N	15.5	17.6	20.3	1.87	0.174
Urinary N	12.5	10.5	11.9	1.25	0.487
Manure N	28.0	28.0	32.2	2.54	0.368
Milk N	11.3	11.0	10.7	1.25	0.932
N excretion, % N intake					
Fecal N	41.4	45.5	44.6	2.00	0.297
Urinary N	33.5	27.7	26.3	2.97	0.178
Manure N	74.9	73.3	70.9	2.95	0.559
Milk N	29.8a	28.6a	23.7b	1.34	0.014
N balance					
N retained, g/d	−1.63	−0.57	2.68	1.59	0.133
N retained, % N intake	−4.69	−1.89	5.44	3.22	0.082

^1^ C = control diet; L = linseed diet; H = hemp seed diet.

**Table 9 animals-11-02717-t009:** Energy balance of the goats fed the experimental diets.

Item	Diet ^1^			SEM	*p*-Value
	C	L	H		
Energy intake, MJ/d	34.5	36.0	41.8	4.01	0.340
Fecal energy, MJ/d	12.1	14.7	15.8	1.63	0.232
Digestible energy, MJ/d	22.3	21.3	26.0	2.53	0.354
Urinary energy, MJ/d	0.909	0.895	0.955	0.098	0.883
Methane energy, MJ/d	2.10	1.90	2.41	0.27	0.396
ME ^2^, MJ/d	19.3	18.5	22.7	2.23	0.349
Heat production, MJ/d	12.2	11.9	13.3	0.94	0.514
Milk energy, MJ/d	5.85	6.25	6.77	1.02	0.772
Retained energy, MJ/d	1.30	0.33	2.62	0.83	0.172
% of energy intake					
Fecal energy	35.3	40.7	37.8	1.59	0.089
Digestible energy	64.7	59.3	62.2	1.59	0.089
Urinary energy	2.64	2.51	2.30	0.18	0.358
Methane energy	6.07	5.27	5.78	0.43	0.415
ME ^2^	56.0	51.6	54.1	1.33	0.094
Heat production	35.4	33.5	32.1	1.49	0.256
Milk energy	16.8	17.4	15.8	1.25	0.648
Retained energy	3.81	0.75	6.16	1.99	0.184
% of ME					
Heat production	63.2	65.1	59.4	3.17	0.405
Milk energy	29.9	33.7	29.0	2.47	0.377
Retained energy	6.61	1.12	11.4	3.67	0.168
Diets energy content					
ME ^2^, MJ/kg DM	10.6	10.3	10.7	0.351	0.646
k_l_ ^3^	0.693	0.668	0.694	0.036	0.836
NE_L_ ^4^, MJ/kg DM	7.38	6.87	7.40	0.479	0.671

^1^ C = control diet; L = linseed diet; H = hemp seed diet. ^2^ ME: metabolizable energy. ^3^ k_l_: efficiency of ME utilization for lactation. ^4^ NE_l_: net energy for lactation.

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
