# Peer review of "Effect of Linseeds and Hemp Seeds on Milk Production, Energy and Nitrogen Balance, and Methane Emissions in the Dairy Goat"

_animals, 2021, doi:10.3390/ani11092717_

Round 1

Reviewer 1 Report

This is a well-written paper that details a large amount of work with hemp and linseed supplementation in goat diets, with the ultimate conclusion that such supplements did not make much difference in milk quality or animal health. I recommend that the article be accepted after some rewriting, and I have a few questions and suggestions.

minor suggestion- drop line 296, it adds nothing to the mss.

What were the sources of the hemp and linseed seeds? There are certainly genetic differences in nutritional/chemical composition of each seed, and this should be mentioned. 

Table 4  why are initial values for the Linseed diet lower than control or hemp in almost all cases, and higher than C or H after 5 hours, whether or not statistically significant?

In many cases in the tables (especially tables 7-9) there are no significant differences-- maybe just present overall values in a combined table. 

Similarly, I suggest combining Results and Discussion to avoid repetition. You can then strengthen or even lengthen the conclusion, although perhaps mention that chemical and nutritional values may vary by seed origin.

Author Response

Dear Reviewer, you can find here in attachment our responses to your comments.

Kind regards,

   G.Matteo Crovetto

Reviewer 2 Report

I suggest that the methane yield and calculations be described in the methodology. VFA analyzes can be expressed in moles/100 moles and determine if there is a significant difference.

Author Response

(The authors gave the same response as above.)

Reviewer 3 Report

 The manuscript reports the "Effect of linseeds and hemp seeds on milk production, energy and nitrogen balance, and methane emissions in the dairy goat". The topic is interesting because it is current and for the implications on human health that may arise from the research.
The work is complete in every part, the introduction is well structured and contains the information to support the purpose.
The M&M are quite detailed and the analyzes performed, missing at what time the milk was collected for the analyzes, start, middle or end of the test? It would be advisable to see a trend in the course of the test of the lipid parameters of the milk, considering that there are few animals.
The results are well expressed, the trend during the test of the various parameters analyzed is missing.
However, I would have reiterated more that diets create a decrease in newly formed saturated fatty acids and for this reason it would be appropriate to see the trend of fatty acids during the test.In my humble opinion, with the integration of those few data, the work can be published on Animals.

Author Response

(The authors gave the same response as above.)
